# REFINE-BY-ALIGN: REFERENCE-GUIDED ARTIFACTS REFINEMENT THROUGH SEMANTIC ALIGNMENT

**Yizhi Song**[1] *    **Liu He**[1]    **Zhifei Zhang**[2]    **Soo Ye Kim**[2]    **He Zhang**[2]    **Wei Xiong**[2]
**Zhe Lin**[2]    **Brian Price**[2]    **Scott Cohen**[2]    **Jianming Zhang**[2]    **Daniel Aliaga**[1]
[1] Purdue University    [2] Adobe Research

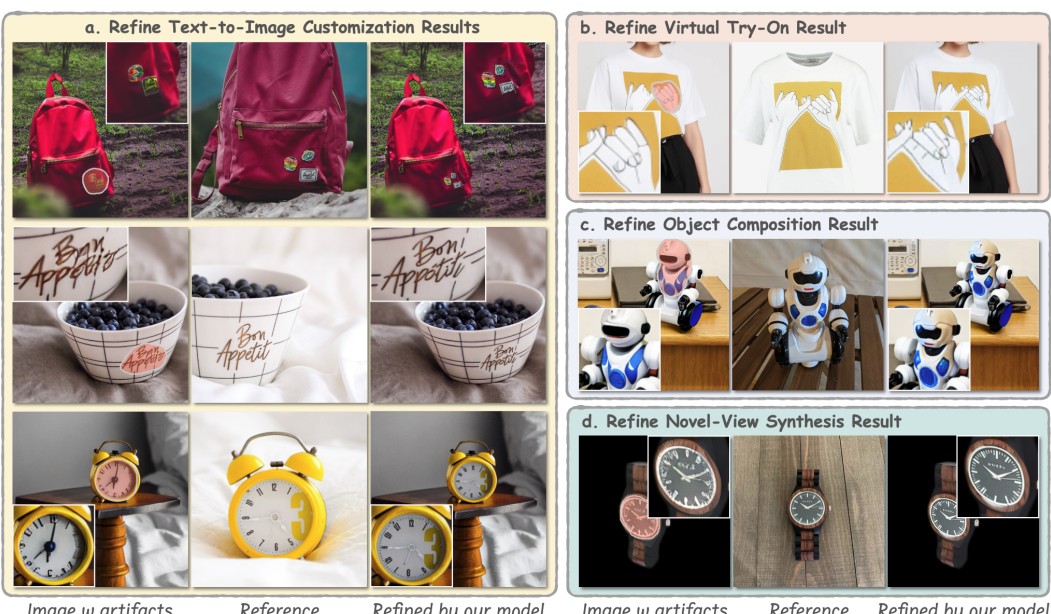

Figure 1: *Refine-by-Align.* Given a generated image (with artifacts), a free-form mask indicating the artifacts region in the generated image, and a high-quality reference image containing important details such as identity logo or font, our model can automatically refine the artifacts in the generated image by leveraging the corresponding details from the reference. The proposed method could benefit various applications (e.g., DreamBooth (Ruiz et al., 2023a) for text-to-image customization, IDM-VTON (Choi et al., 2024) for virtual try-on, AnyDoor (Chen et al., 2023) for object composition, and Zero 1-to-3++ (Shi et al., 2023b) for novel view synthesis.

## ABSTRACT

Personalized image generation has emerged from the recent advancements in generative models. However, these generated personalized images often suffer from localized artifacts such as incorrect logos, reducing fidelity and fine-grained identity details of the generated results. Furthermore, there is little prior work tackling this problem. To help improve these identity details in the personalized image generation, we introduce a new task: *reference-guided artifacts refinement*. We present **Refine-by-Align**, a first-of-its-kind model that employs a diffusion-based framework to address this challenge. Our model consists of two stages: **Alignment Stage** and **Refinement Stage**, which share weights of a unified neural network model. Given a generated image, a masked artifact region, and a reference image, the alignment stage identifies and extracts the corresponding regional features in the reference, which are then used by the refinement stage to fix the artifacts. Our model-agnostic pipeline requires no test-time tuning or optimization. It automatically enhances image fidelity and reference identity in the generated image, generalizing well to existing models on various tasks including but not

---

*This work was done while the author was an intern at Adobe Research.

limited to customization, generative compositing, view synthesis, and virtual try-on. Extensive experiments and comparisons demonstrate that our pipeline greatly pushes the boundary of fine details in the image synthesis models. Project page: https://song630.github.io/Refine-by-Align-Project-Page/

# 1 INTRODUCTION

Generative models (Goodfellow et al., 2014; Karras et al., 2019; He & Aliaga, 2024; 2023) for image synthesis (Ho et al., 2020; Rombach et al., 2022; Peebles & Xie, 2023; Podell et al., 2023; Luo et al., 2023b) have made significant advancement. Moreover, the traditional task of reference-guided image generation (Sheng et al., 2022; 2023; 2024) has been enabled by recent diffusion models (DM), where a text and/or visual prompt is provided and the subject object is generated in a specified novel context. This ability has been widely applied to applications such as subject customization (Ruiz et al., 2023a), object composition (Chen et al., 2023), novel view synthesis (Shi et al., 2023b) and virtual try-on (Choi et al., 2024). While these works seek generation in a single step, in practice undesired blemishes, detail omissions, and blurriness may occur in the generated images. These localized unpleasant anomalies are typically called *artifacts* as perceived by human eyes (e.g., "artifacts" row in Figure 1 (Zhang et al., 2023b)). The artifacts reduce image fidelity and the overall prompt-alignment quality of the synthesized images. Thus, a localized refinement tool to remove or reduce artifacts is beneficial.

Recently, a few limited approaches to artifact detection and refinement have been presented. PAL (Zhang et al., 2023b) presents an early work in this area that trains an artifact detection model in a supervised end-to-end manner using input images with artifacts and corresponding ground truth images. The detected artifacts can be partially removed using a pre-trained image inpainting tool. As PAL identifies, the artifacts are typically very small and irregular-shaped (Appendix Fig. 8), which further complicates the refinement process. PAL ameliorates artifacts but struggles with diversity of artifact refinement. Lack of diversity is also observed in RealisHuman (Wang et al., 2024a) that focuses on artifacts in human hands and faces, and the SynArtifact (Cao et al., 2024) using vision-language model and reinforcement learning for artifact annotation and removal. In general, none of these methods are able to provide a controllable and predictable artifact refinement output with free-form support automatically to precisely preserve the original identity details.

Our main approach is to leverage the identity detail info in the reference-image to guide the refine the artifacts. This provides a locally-controllable output (i.e., we specify the desired refinement), works for arbitrarily-shaped artifacts, preserves the identity and background in the provided generated image, and is applicable to multiple image generation approaches. As shown in Fig. 1, we provide *reference-guided artifacts refinement*: Given a generated image with marked artifact regions, and a reference image (containing a reference object), our model refines the artifacts by transferring corresponding details from the reference image to the artifact regions in the originally generated image. Our reference-guided approach shares insights with many reference-based image customiza-

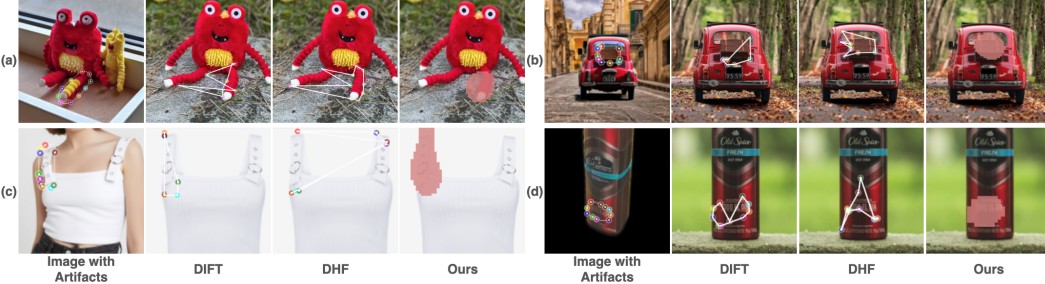

Figure 2: Comparisons of our region-matching method with keypoint matching. We utilize DIFT (Tang et al., 2023) and DHF (Luo et al., 2023a) to perform keypoint matching from the artifacts region (10 points are sampled along the artifacts contour) to the reference. DIFT and DHF often fail to find the accurate corresponding region; in addition, they have trouble in distinguishing between repeating patterns such as (a)(c). In contrast, our method is more robust. The results demonstrate that artifacts alignment is a non-trivial process.

tion models (Song et al., 2023; Chen et al., 2023; Yang et al., 2023), but those models overlooked the accurate region-alignment between reference image and artifact region. In Fig. 2, SOTA models specialized on finding correspondences may fail on key point matching for arbitrary-shaped regions.

Our approach proceeds in two stages (Fig. 3). (1) **Alignment Stage**: we design a novel alignment algorithm (Algorithm 1) for arbitrarily-shaped regional feature matching which utilizes the artifact regions to query corresponding features from the reference image. In particular, the best match is obtained by maximizing the spatial correlation between DM-captured cross-attention maps of both artifact region and the reference image. (2) **Refinement Stage**: we train a diffusion model to use the best matched features from the alignment stage to refine the artifact region and preserve identity. Our model is trained in a self-supervised scheme and we demonstrate its application to artifacts generated by various generative models. Quantitative and qualitative comparisons (i.e., using several well-established metrics and a user study; Sec. 4.2, Sec. 4.3) show that in terms of detail and appearance preservation our model outperforms all six baseline models (Paint-by-Example (Yang et al., 2023), ObjectStitch (Song et al., 2023), AnyDoor (Chen et al., 2023), PAL (Zhang et al., 2023b), Cross-Image Attention (Alaluf et al., 2024) and MimicBrush (Chen et al., 2024)).

Our contributions can be summarized as follows:

- A first-of-its-kind generative artifacts refinement framework supporting control via reference image specification, identity preservation, arbitrary artifact shapes, and good fidelity.
- A novel artifacts matching algorithm which matches arbitrarily-shaped artifacts to corresponding patterns in the reference image.
- An effective reference-guided refinement strategy that ameliorates artifacts in a provided generated image.
- A comprehensive benchmark, *GenArtifactBench*, consisting of artifacts generated by several well-known models, reference images, and dense human annotations which can serve to evaluate future efforts in this area.

## 2 RELATED WORK

### 2.1 REFERENCE-GUIDED IMAGE EDITING

Reference-guided image editing has been a traditional task that has various applications, including reference-guided super resolution (Zhang et al., 2019; Jiang et al., 2022), and guided inpainting or outpainting (Zhou et al., 2021; Tang et al., 2024a). With the significant advancements in diffusion models (DM) (Ho et al., 2020; Sohl-Dickstein et al., 2015; Song & Ermon, 2019; Rombach et al., 2022; Ho & Salimans, 2022; Peebles & Xie, 2023; He et al., 2023; Huang et al., 2023) in text-to-image (T2I) synthesis, there have been many works on subject-driven image editing. The notion of using an additional reference image to guide image editing (e.g., (Ruiz et al., 2023a; Kawar et al., 2023)) has led to a series of techniques (Ruiz et al., 2023b; Gal et al., 2022; Shi et al., 2023a; Kumari et al., 2023; Liu et al., 2023b) using optimization to learn concepts. In spite of their high-fidelity editing results, they usually require inference time fine-tuning or multiple subject images. Another branch of works (Yang et al., 2023; Song et al., 2023; Chen et al., 2023; Zhang et al., 2024b; Li et al., 2024; Xiong et al., 2024) replace the text embedding of T2I models with image embedding based on CLIP or DINOv2 (Radford et al., 2021; Oquab et al., 2023; Song et al., 2024), which are tuning-free. Subsequent works have extended the applications to image compositing (Lu et al., 2023; Wang et al., 2024b; Avrahami et al., 2022; Sarukkai et al., 2024; Meng et al., 2021), novel view synthesis (Liu et al., 2023a; Shi et al., 2023b; Liu et al., 2024b), instruction-guided editing (Pan et al., 2023; Yu et al., 2024; Tang et al., 2022) via Large Language Models (Liu et al., 2024a; Zhang et al., 2024a) or Vision-Language Models (He et al., 2024; Hua et al., 2024b;a; Tang et al., 2024c;b), and subject swapping (Gu et al., 2023; 2024). However, they all face a critical challenge of controllability and identity preservation of the original object.

### 2.2 CORRESPONDENCE MATCHING

Traditional methods use carefully-designed features (Bay et al., 2006; Lowe, 2004; Rublee et al., 2011) or learning-based approaches (Aberman et al., 2018; Rocco et al., 2018; Seo et al., 2018;

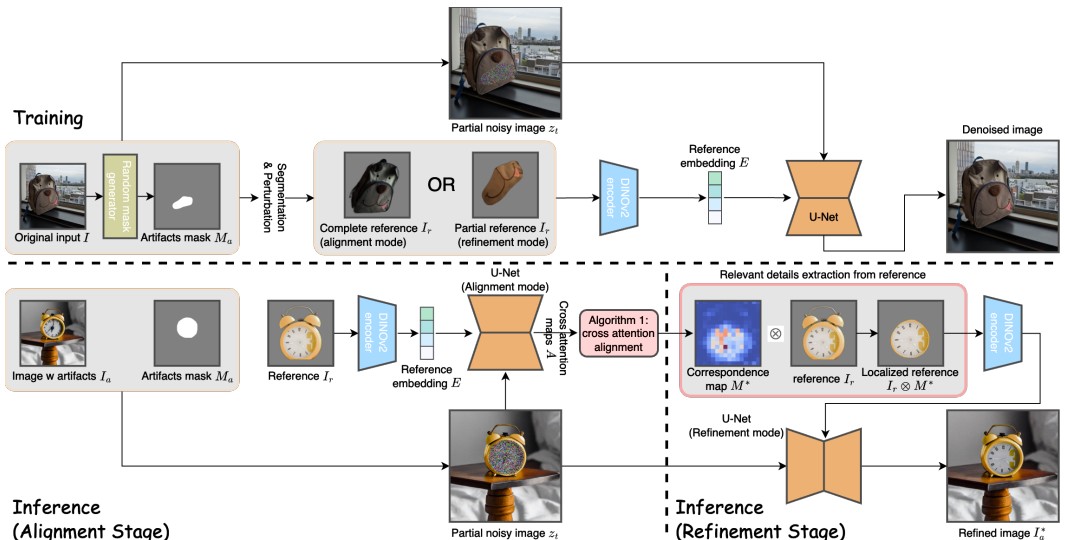

Figure 3: **Overview of our framework.** *Top*: During training, we train a DM for object completion, guided by a reference image $I_r$. In alignment mode, the reference is a complete object, so the model learns to locate the relevant region from the reference for object completion, thus maximizing the spatial correlation in attention maps. In refinement mode, this region is directly provided as reference. *Bottom*: During inference, the inputs include a generated image $I_a$ with the artifacts marked as $M_a$, and a reference object $I_r$. In the alignment stage, we perform **cross-attention alignment** algorithm (see Alg. 1 and Fig. 4) to find the correspondence map $M^*$. In the refinement stage, $M^*$ is used to find the region in $I_r$ that corresponds to artifacts, which guides refining to $I_a$.

Simonyan et al., 2014) to establish correspondences. Recent work has shifted towards adapting diffusion models. Tang et al. (2023); Luo et al. (2023a); Zhang et al. (2023a); Hedlin et al. (2023) leverage the features maps or embeddings of pretrained DMs to predict correspondences. Appearance transfer (Go et al., 2024; Chen et al., 2024; Alaluf et al., 2024) is a downstream application of such task. However, they still suffer from the loss of fine-grained identity details from the reference.

## 2.3 LOCALIZATION AND REFINEMENT OF ARTIFACTS

It is challenging for the state-of-the-art models to capture intricate details, such as object textures and human hands. In Zhang et al. (2023b), these artifacts are defined as *implausible content or display of unpleasant artifacts in specific regions of the image*. Although artifacts are commonly observed even in the leading generative models, few works have explored the topic of artifacts detection and refinement. Recently, Zhang et al. (2023b) curate a human-annotated dataset for artifact segmentation, train a segmentation model for artifact localization and a zoom-in inpainting pipeline for refinement. Cao et al. (2024) classify the artifacts and fine-tune T2I DMs to reduce artifacts. Specifically in handling artifacts on human body, Wang et al. (2024a) propose an approach to refine artifacts in faces and hands. Concurrently, Chen et al. (2024) propose MimicBrush for appearance transfer, which is a potential use for artifact refinement. However, none of these works are designed to perform reference-guided artifacts refinement, thus lacks control. For the first time, we provide a solution for guided artifacts refinement with fine-grained control.

## 3 METHOD

The proposed artifacts refinement framework, Refine-by-Align , is shown in Fig. 3. Formally, we define the task of artifacts refinement conditioned on a reference image as following: Let $I_a \in \mathbb{R}^{H \times W \times 3}$ be an image with artifacts generated by any reference-guided generation model, $M_a \in \mathbb{R}^{H \times W}$ be a user provided artifacts mask, $I_r \in \mathbb{R}^{H \times W \times 3}$ be the segmented reference object image; we train a Diffusion Model (DM)-based refinement model $\mathcal{R}$ to generate a refined image $I_a^*$:

$$I_a^* = \mathcal{R}(I_a, M_a, I_r) \in \mathbb{R}^{H \times W \times 3} \tag{1}$$

Ideally in $I_a^*$, the refined area $I_a^* \otimes M_a$ should be consistent with the background and preserve the identity from the reference object $I_r$; the whole image $I_a^*$ should appear natural and artifacts-free.

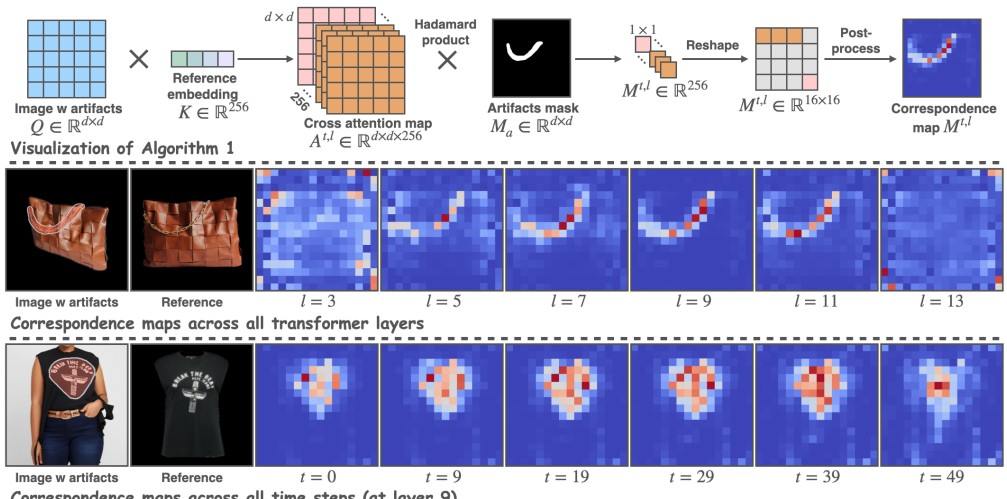

Figure 4: *Top*: Visualization of our cross-attention alignment algorithm. The artifacts mask is used to extract the spatial correlations between the artifacts and the reference; the output of this algorithm, the correspondence map, indicates the region in the reference that corresponds to the artifacts area. *Middle and Bottom*: Correspondence maps across different transformer layers and timesteps.

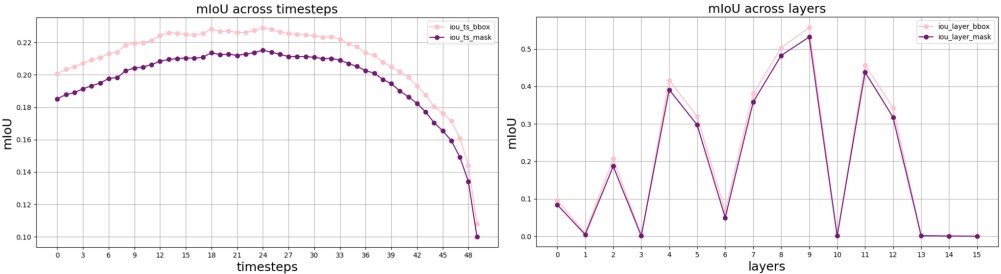

Figure 5: Running the cross-attention alignment algorithm on GenArtifactBench to find the best combination of timestep $t$ and transformer layer $l$. *Left*: mIoU across all timesteps, averaged over all layers and images; *Right*: mIoU across all layers, averaged over all timesteps and images.

However, directly using $I_r$ as the visual guidance may cause significant degradation. Hence, an **optimal correspondence mask** $M^*$ is necessary for an optimized input reference as $I_r \otimes M^*$ for better guidance. To tackle this task, we created a two stage approach $\mathcal{R}$: the **Alignment Stage** and the **Refinement Stage**, which share the weights of a single unified neural network model. In the alignment stage, the artifacts region $I_a \otimes M_a$ is used as the query to localize the optimal correspondence region mask $M^*$ in $I_r$. In the refinement stage, $I_r \otimes M^*$ is fed into DINOv2 (Oquab et al., 2023) encoder to extract expressive visual features, guiding the local generation process to eliminate the artifacts while ensuring identity preservation.

Our design is motivated by two key observations:

- Refine by localization: As summarized in (Zhang et al., 2023b), generative artifacts occurs more frequently around tiny object details such as logos, texts and other complex textures. However, these areas have higher fidelity when such details are prominent in the image. Based on this observation, we assume that the fidelity can be improved by performing a local generation guided by a local region from the reference.

- Align via cross-attention: As demonstrated in Hertz et al. (2022), the appearance of the generated image depends on the interaction between the pixels to the text embedding, which occurs in cross attention layers. Similarly, when replacing the text embedding with image embedding, we assume that spatial correspondence exists between the generated image and the image embedding.

Based on the above observations, we review cross-attention and prove that the spatial correspondence exists in Sec. 3.2; the algorithm to align $I_a \otimes M_a$ to the corresponding local region in $I_r$ is explained in Sec. 3.3. Sec. 3.4 describes the two training modes. Sec. 3.5 describes the inference.

---

**Algorithm 1** Optimal Cross-Attention Alignment (refer to Fig. 4 for visualization)

---

**Input**: Target image $I$, resized artifacts mask $M_a \in \mathbb{R}^{d \times d}$, reference image $I_r$, DINOv2 encoder $\phi$, the refinement model $\mathcal{R}$, the ground truth correspondence mask $M_{gt}$ on $I_r$.
**Parameter**: The diffusion time steps $T$, total number of transformer block $L$ in $\mathcal{R}$, the noisy image resolution $d$ in latent space.
**Output**: Optimal correspondence mask $M^*$ (for $I_r$).

1: $E \leftarrow \text{MLP}(\phi(I_r)); E \in \mathbb{R}^{256 \times 768}$ . . . . . . . . . . . . . . . . . . . . . . . . . . . . # Get reference embedding
2: $z_T \sim \mathcal{N}(0, I)$; Zero array $\Gamma \in \mathbb{R}^{T \times L}$
3: **for** $t = T - 1, \cdots, 0$ **do**
4:     **for** $l = 0, \cdots, L - 1$ **do**
5:        $A^{t,l} \leftarrow \mathcal{R}(z_t, t, E); A^{t,l} \in \mathbb{R}^{(d \times d) \times 256}$ . . . . . # Extract cross-attention map at $l$-th block
6:        $M^{t,l} = \sum_{i,j}(M_a \circ A^{t,l}[i,j,:])$ . . . . . . . . . . . . . . . . . . . . # 2D correspondence mask on $I_r$
7:        $\Gamma_{t,l} \leftarrow \text{mIoU}(M^{t,l}, M_{gt})$ . . . . . . . . . . . . . . . . . . . . . . . . . . . . . . . . . . . # Calculate mIoU
8:     **end for**
9: **end for**
10: $t^*, l^* \leftarrow \arg \max_{t,l}(\Gamma)$ . . . . . . . . . . . . . . . . . . . . . . . . . . . . . . . . . . . . . . . . . . # Find optimal $t$ and $l$
11: **return** $M^* \leftarrow M^{t^*,l^*}$

---

### 3.1 DIFFUSION MODEL

We leverage the architecture of AnyDoor (Chen et al., 2023) and IMPRINT (Song et al., 2024) as the backbone of our refinement network. It is based on the Stable Diffusion (Rombach et al., 2022) model, which contains three major components: the variational autoencoder (VAE) to code images into latent embeddings $z$, the U-Net backbone $\mathcal{R}$ parameterized by $\theta$ for sequentially denoising diffusion steps on $z$ by Gaussian noise $\epsilon$, and a text encoder for guidance injection. We replace the text encoder by a vision encoder $\phi$ based on a pretrained DINOv2 (Oquab et al., 2023) to enable visual prompt guidance by a reference image $I_r \in \mathbb{R}^{H \times W \times 3}$ of a subject. In particular, the conditional embedding $\phi(I_r)$ will be utilized for optimization of loss functions on artifact refinement:

$$\mathcal{L}_{artifact} = \mathbb{E}_{z,I_r,\varepsilon \sim \mathcal{N}(0,I),t} \left[ \|\varepsilon - \mathcal{R}_\theta(z_t, t, \phi(I_r))\|_2^2 \right] \tag{2}$$

where $z_t$ is the latent embedding at time step $t$. $\mathcal{R}$ is trained to iteratively denoise $z_T$ to $z_0$.

### 3.2 SPATIAL CORRESPONDENCE IN ENCODER-BASED CUSTOMIZATION MODELS

Our vision encoder $\phi$ based on DINOv2 is employed to extract the image embedding from the reference object, which is then fed to the cross attention layers of the U-Net backbone. Its ViT backbone divides the input image into $16 \times 16$ square patches, thus the encoded $\phi(I_r)$ is a sequence of 256 tokens and can be mapped back to 2D space with original spatial information as followed:

$$E = \phi(I_r); E \in \mathbb{R}^{256 \times d^\phi} \tag{3}$$

In each attention layer, $\phi(I_r)$ is projected to the key $k = \psi_k(E)$ and the value $v = \psi_v(E)$ by the linear projections $\psi_k$ and $\psi_v$. Meanwhile, the noisy image encoded by VAE to latent embedding $z_t$ is projected to the query $q = \psi_q(z_t)$. Specifically, $z_t \in \mathbb{R}^{d \times c}$, where $d \in \{64, 32, 16, 8\}$ is the resolution of the extracted image features at different depth of layers, and $c$ is the number of channels. $z_t$ also contains spatial information. Altogether, cross attention map $A$ is calculated as:

$$A = softmax(qk^T); A \in \mathbb{R}^{(d \times d) \times 256} \tag{4}$$

Intuitively, $A$ measures the similarity between $q$ and $k$, and $A_{[x,y,k]}$ stores the amount of information flow from the $k$th reference token to the latent pixel at $(x, y)$ (Xiao et al., 2023). Therefore, through cross attention, $A$ effectively encodes the interaction between the noisy image and the reference embedding connected by spatial correlations.

### 3.3 THE ALIGNMENT ALGORITHM

Although the cross attention map $A$ encodes the spatial correlations between $I_a$ and $I_r$, two challenges remain in identifying $M^*$: (1) given $A$, accurately locating the region in the reference image

that corresponds to the free-form artifacts $I_a \otimes M_a$; (2) finding the optimal correspondence map $\boldsymbol{M}^*$ from the numerous maps $\boldsymbol{M}^{t,l}$ of all possible combinations of diffusion timesteps $t$ and transformer layers $l$. Aiming at these two challenges, we present our **cross-attention alignment** algorithm, shown as pseudo code in Algorithm 1, and visualized in the top part of Fig. 4.

**Optimal Cross-Attention Alignment**

Since $z_t$ is obtained by encoding a partially noisy version of $I_a$ (see Sec. 3.4), each pixel $I_a[i, j]$ can be mapped to its spatial correspondence in $z_t$. Given that $\boldsymbol{A}$ encodes the correlation between the noisy image $z_t$ and the reference tokens $\boldsymbol{E}$ (discussed in Sec. 3.2), it can be concluded that any pixel $I_a[i, j]$ can find its match in $\boldsymbol{E}$ via $\boldsymbol{A}$.

For the correspondence map of $I_r$ at diffusion timestep $t$ and transformer layer $l$, we accomplish this by aggregating all pixels belonging to the artifacts:

$$\boldsymbol{M}^{t,l} = \sum_{i,j}(M_a \circ \boldsymbol{A}[i,j,:]) \tag{5}$$

where the Hadamard product is calculated between the artifacts mask and the cross-attention map. Intuitively, the 2D correspondence map $\boldsymbol{M}^{t,l}$ measures the similarity between all reference tokens and the artifacts pixels. To obtain the optimal $\boldsymbol{M}^*$, some post-processing needs to be applied on $\boldsymbol{M}^{t,l}$. Refer to Sec. A.4 for more details.

A grid search over all possible $t, l$ values is performed on our proposed benchmark (see Sec. 4.1) and evaluated using mIoU. The results are shown in Fig. 5, demonstrating that the optimum combination is $t = 24$ and $l = 9$. However, during test time, we choose $t = 0, l = 9$ to accelerate the inference. Detailed comparisons can be found in Sec. 4.4.

## 3.4 TRAINING

For training, we implement two modes sharing weights of the same model, where the only difference lies in the inputs. In both modes, we train the U-Net, and the MLP connecting DINOv2 and U-Net.

**Alignment Mode**

To maximize the spatial correlations contained in the cross-attention maps, we design a fully self-supervised training scheme. The idea behind the construction of training pairs is: *Use a complete reference object to guide object completion, forcing the model to learn to identify the corresponding local region from the reference.*

We use Pixabay (Song et al., 2023) with panoptic segmentation labels as the training dataset. Following the aforementioned idea, given a Pixabay image $I_o$ and an object mask $M_o$, we start by applying a random mask generator $\mathcal{G}$ to sample a free-form artifacts mask $M_a = \mathcal{G}(M_o)$ within $M_o$. We then design color perturbation $\mathcal{S}$ and affine transformations $\mathcal{T}$ on $I_o \otimes M_o$ to simulate the lighting and view changes in the real world: $I_r = \mathcal{S}(\mathcal{T}(I_o \otimes M_o))$. In this mode, the artifacts image $I_a = I_o \times (1 - M_o)$ and the perturbed reference $I_r$ are used as the inputs, and $I_o$ is the target.

**Refinement Mode**

In this mode, it is assumed that the local region corresponding to the artifacts is already given from the original reference, which is used as $I_r$ to guide the inpainting of the incomplete object.

Our training data consists of Pixabay and MVObj, a manually annotated dataset where an object appears in multiple images with different contexts and views. MVObj is added since our perturbations $\mathcal{S}, \mathcal{T}$ are not sufficient to simulate non-rigid or 3D pose changes. 1) When dealing with a Pixabay image, we obtain the partial reference using the artifacts mask: $I_r = \mathcal{S}(\mathcal{T}(I_o \otimes M_a))$; 2) when processing a pair of MVObj images $(I_{o1}, I_{o2}, M_{o1}, M_{o2})$, $I_{o1} \otimes erode(M_{o1})$ is used as the reference, and $I_{o2} \otimes 1 - (erode(M_{o2}))$ is the input artifacts image $I_a$, where the mask is eroded.

We then adopt the *zoom-in inpainting strategy* from Zhang et al. (2023b), cropping around $I_a$ and perform inpainting on the zoom-in patch.

## 3.5 INFERENCE

Inference is performed in two stages, corresponding to the above two modes. In the alignment stage, given $I_a, M_a, I_r$, our fine-tuned model $\mathcal{R}$ follows Alg. 1 and produces the correspondence map

Table 1: **Quantitative Comparison** with the baseline models on *GenArtifactBench*. PbE (Paint-by-Example) and OS (ObjectStitch) are finetuned on our training set; the local regions (instead of the complete reference) are provided to PbE, OS and AnyDoor. Our method outperforms the other baselines in fidelity. Since CLIP-T only captures high-level semantics, it cannot accurately measure the detail preservation. We show additional comparisons in Tab. 2 and Fig. 6.

| Categories | Methods | CLIP-T ↑ | CLIP-I ↑ | DINO-I ↑ |
|---|---|---|---|---|
| Reference-guided inpainting | Paint-by-Example* | 23.5938 | 80.7500 | 58.7625 |
| | ObjectStitch* | 24.4844 | 83.9375 | 72.6152 |
| | AnyDoor | 25.0625 | 83.4375 | 71.3398 |
| Text-guided inpainting | PAL | **25.8906** | 81.5000 | 53.8117 |
| Appearance transfer | Cross-Image Attention | 23.2500 | 80.2500 | 56.7892 |
| | MimicBrush | 25.0156 | 85.0625 | 67.6194 |
| | **Ours** | 25.4063 | **86.6250** | **75.3135** |

$M^*$. Note that the grid search is skipped and $\mathcal{R}$ only proceeds one timestep, directly producing the optimal correspondence map $M^*$ at timestep 0 and layer 9. The local region that corresponds to the artifacts can be extracted from the reference as $I_r \otimes M^*$.

In the refinement stage, $I_r \otimes M^*$ is provided as the reference through DINOv2, guiding the refinement of $I_a$ via inpainting the artifacts region.

## 4 EXPERIMENT

### 4.1 EVALUATION BENCHMARK

**Dataset.** To provide insight on the appearance of generative artifacts and an effective evaluation of our artifacts refinement approach, we present **GenArtifactBench**, the first benchmark for reference-guided artifacts refinement (refer to the Appendix for examples), featuring:

- We generate images using four popular models: DreamBooth (Ruiz et al., 2023a), Zero123++ (Shi et al., 2023b), AnyDoor (Chen et al., 2023) and IDM-VTON (Choi et al., 2024), covering real-world applications of T2I customization, view synthesis, compositing and virtual try-on. Diverse artifacts are shown in these images.
- We collect 146 generated images with notable artifacts, paired with 146 reference images of more than 40 objects (from DreamBooth and Pixabay) and 27 garments (from Choi et al. (2021)).
- Human annotation of the artifacts, and the corresponding regions from the reference images.

**Metrics.** To measure the semantics and identity preservation of refined object, we utilize CLIP-Image Score (Radford et al., 2021) and DINO Score (Oquab et al., 2023) to compute the similarity between the refined region of the generated image and its corresponding region from the reference. When calculating CLIP-Text Score, BLIP2 (Li et al., 2023) is used to generate captions. Since a measurement of the overall quality is absent, we further conduct a user study.

**Parameters.** $t = 0$ and $l = 9$ are used in all the comparisons below.

### 4.2 QUANTITATIVE COMPARISONS

To demonstrate the effectiveness of our model, we compare our model with 6 baseline models (PbE or Paint-by-Example (Yang et al., 2023), OS or ObjectStitch (Song et al., 2023), AnyDoor (Chen et al., 2023), PAL (Zhang et al., 2023b), Cross-Image Attention (Alaluf et al., 2024) and MimicBrush (Chen et al., 2024), which is a *concurrent* work) on *GenArtifactBench* using the aforementioned metrics. For fair comparison, PbE and OS are fine-tuned on our training dataset.

Quantitative comparisons are shown in Tab. 1, where the baselines are categorized into three classes. Note that we provide the accurate reference regions for PbE, OS and AnyDoor, greatly reducing the difficulty for these models. In terms of semantics preservation measured by CLIP-T, PAL shows a slight advantage over our model; however, as illustrated in Fig.6, it fails to gain fine-grained control over the details in local editing. This is because PAL is built on SDXL (Podell et al., 2023), which is only performing inpainting following a high-level text description. CLIP-I and DINO score are

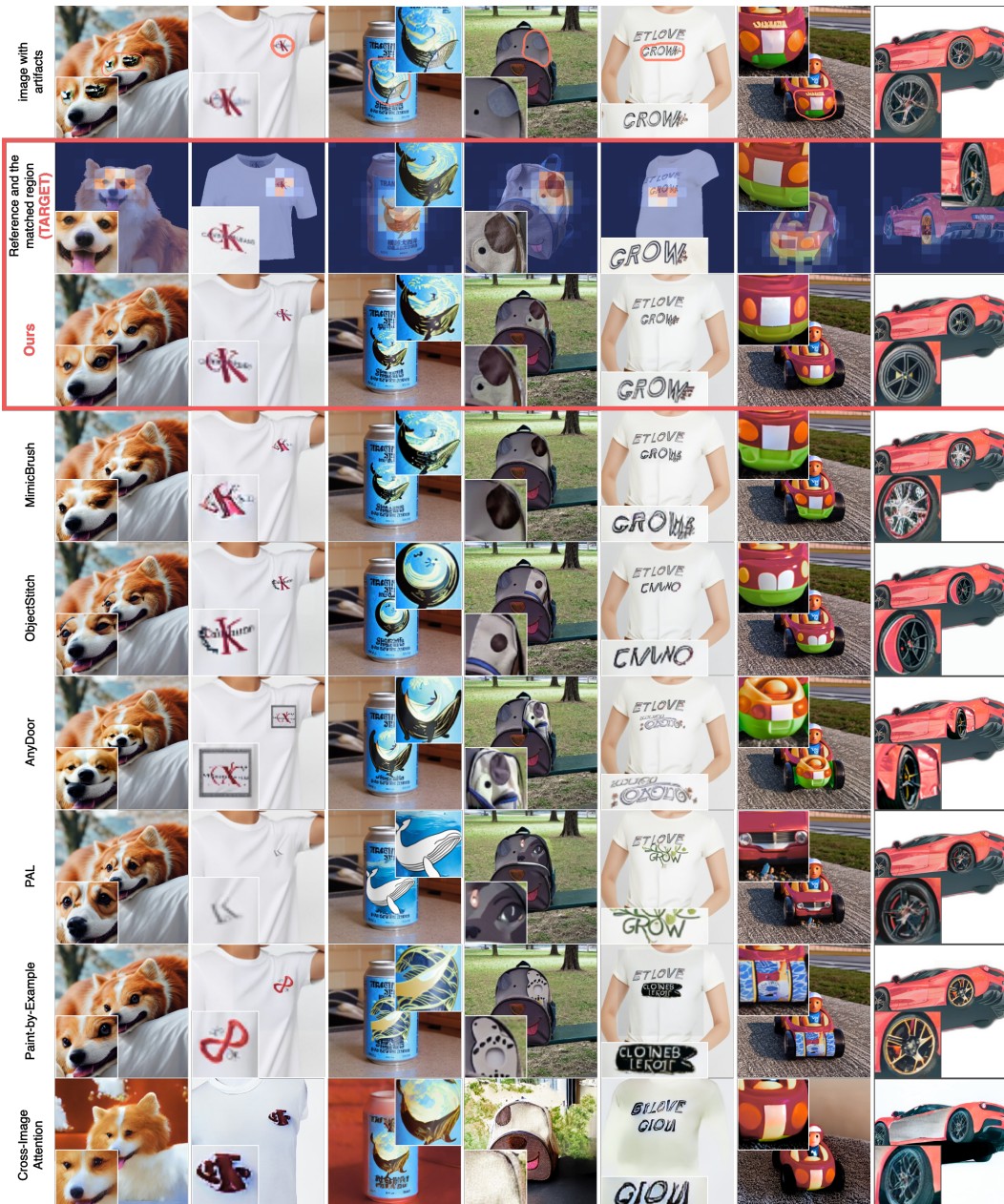

Figure 6: **Qualitative comparisons.** *Zoom in* to view details. Note that the accurate reference regions corresponding to the artifacts (not the complete reference) are provided to PbE, OS and AnyDoor. In the second row of the references, we overlay the correspondence maps on them. Compared with the baselines, our model not only preserves identity (most similar to the second row), but also generate smooth and natural results where artifacts are significantly reduced.

computed only over the masked region. Our model outperforms all baseline models in details and appearance preservation. Owning to our correspondence matching strategy, irrelevant feature is removed and consequently the identity is significantly improved.

To take human perception into account as well as adding quality metrics, we conduct a user study on Amazon Mechanical Turk. We design two questions to assess identity preservation and generation realism, respectively. In each question, side-by-side comparisons are presented to the user: our result alongside one selected from a baseline. It is important to note that the reference images remain hidden from the user when assessing quality. Each question has 240 comparisons, and 720 votes are collected from more than 150 workers. The user preference is reported in Tab. 2. The preference rate demonstrates the superiority of our model over all baselines in both fidelity and realism.

Table 2: **User study.** Results are user preference win rate (%). Two questions are designed to measure identity preserving and realism; in each question the user is presented side-by-side results of our model and a random baseline. For fairness, PbE and OS are fine-tuned on our training dataset.

| Identity ↑ | | | | Realism ↑ | | |
|---|---|---|---|---|---|---|
| Ours | **70.83** | Paint-by-Example* | 29.17 | Ours | **71.67** | Paint-by-Example* | 28.33 |
| Ours | **55.83** | ObjectStitch* | 44.17 | Ours | **56.67** | ObjectStitch* | 43.33 |
| Ours | **57.50** | AnyDoor | 42.50 | Ours | **60.00** | AnyDoor | 40.00 |
| Ours | **74.17** | Cross-Image Attention | 25.83 | Ours | **83.33** | Cross-Image Attention | 16.67 |
| Ours | **61.67** | PAL | 38.33 | Ours | **71.67** | PAL | 28.33 |
| Ours | **60.00** | MimicBrush | 40.00 | Ours | **59.17** | MimicBrush | 40.83 |

## 4.3 QUALITATIVE RESULTS

Qualitative comparisons with the baseline models are shown in Fig. 6. The correspondence maps obtained by our model are overlaid on the reference image, highlighting the local regions that match the artifacts. When testing on PbE, OS and AnyDoor, these local regions (instead of the complete reference objects) are directly provided as input. In particular, PbE, OS and PAL struggle in preserving the finer details from the reference. This is especially the case for complex patterns since they only have high-level semantic control over the generation. AnyDoor can capture identity but fails to generate smooth transition areas. MimicBrush, a concurrent work, is designed for reference-guided local editing which shows superiorities over the other baselines; however, our model outperforms MimicBrush in both realism and fidelity.

## 4.4 ABLATION STUDY

**Diffusion timestep and transformer layer.** To evaluate the accuracy of the correspondence maps $M_{t,l}$, we ablate on all timesteps and layers $t \in \{0, 1, ..., 49\}; l \in \{0, 1, ..., 15\}$, and compute the mIoU over all images from *GenArtifactBench*, which is shown in Fig. 4. We also show a 2D mIoU figure on all combinations of $t$ and $l$ in the Appendix (Fig. 7), visualizing $\Gamma \in \mathbb{R}^{T \times L}$. To balance between efficiency and accuracy, we choose $t = 0, l = 9$.

**Comparisons with keypoint matching.** We compare our alignment algorithm to keypoint matching performance by two correspondence matching methods: DIFT (Tang et al., 2023) and DHF (Luo et al., 2023a). Visual results are in Fig. 2. While baselines struggle with repeating or irregular patterns, our alignment algorithm is more robust, with only a single query to locate the target region. Furthermore, the alignment and refinement stages are integrated into a unified model.

**Ablation on model design.** To demonstrate the effectiveness of our architecture design, we compare our full model with two settings (Tab. 3). 1) the model is only trained in the alignment mode, where a complete object is used as the reference. When the irrelevant patterns have been removed from the reference, identity preservation is significantly improved; 2) DINOv2 encoder is replaced by CLIP. The comparison proves that CLIP fails to encode low-level details, thus losing identity.

## 5 LIMITATIONS, CONCLUSION AND FUTURE WORK

We have shown a novel approach to artifact refinement via region alignment and reference-guided generation. Our approach makes use of a high-quality reference image to provide a predictable and controllable refinement output, that also preserves identity and transfers the details from the reference image to the input image containing artifacts. Our method has been compared to several baseline methods and has shown

Table 3: **Ablation Study** on two model designs. 1) using a model which is only trained in the alignment mode to perform single stage artifacts refinement; 2) DINOv2 is replaced by CLIP encoder.

| | CLIP-T ↑ | CLIP-I ↑ | DINO ↑ |
|---|---|---|---|
| Single-stage | 24.4375 | 84.5000 | 71.7609 |
| CLIP-encoder | 24.3750 | 84.3750 | 70.7714 |
| **Full** | **25.4063** | **86.6250** | **75.3135** |

shown consistently superior performance. As limitations, first, our method does not always work well when there is a large disparity between the objects in the original input image and in the reference. Second, the accuracy of our alignment method is limited by the number of vision tokens, where only $16 \times 16$ patches are used to represent an image. As future work, we would like to extend our method to automate artifact detection and to incorporate the use of multiple reference images and text-descriptions so as to obtain blended outputs.

ACKNOWLEDGMENTS

This research is partially funded by NSF Grant 2107096, 1835739 and 2411273.

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
