## A   APPENDIX

### A.1   ABLATION STUDY ON TIMESTEP AND LAYER

Since Fig.4 in the main paper only shows the effect of either changing the timestep $t$ or changing the layer $l$, we also show a 2D mIoU map on all possible combinations of $t$ and $l$, visualizing $\Gamma \in \mathbb{R}^{T \times L}$ in Fig. 7. It can be concluded that the information of spatial correlation encoded in a layer is similar across all timesteps; and the information stored in different layers varies dramatically, where the most precise correlation is mirrored in layer 9.

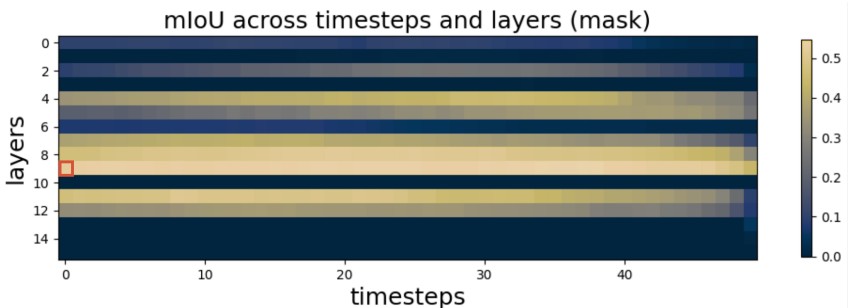

Figure 7: Grid-search results of all transformer layers and diffusion time steps. The 2D heatmap shows mIoU over all possible combinations of the parameters. The highlighted block is our chosen setting for running the inference.

### A.2   ANALYSIS OF PERCEPTUAL ARTIFACTS

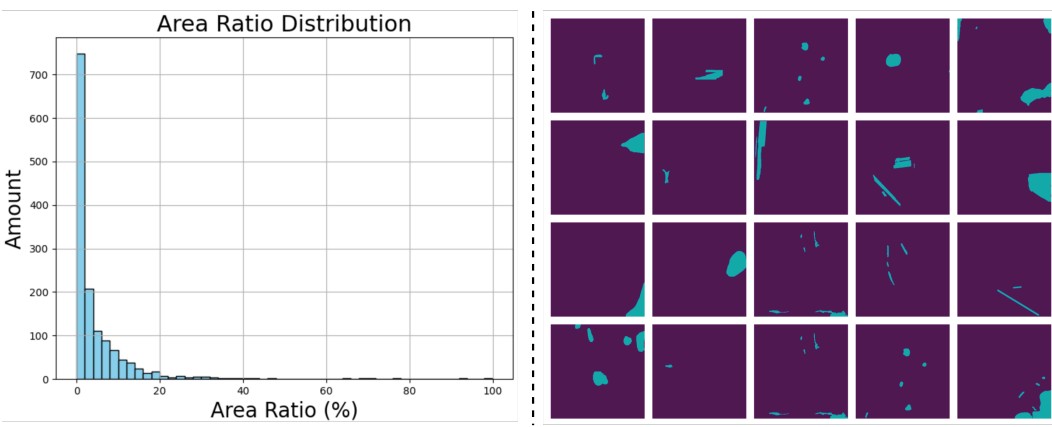

Figure 8: Statistics of PAL artifacts dataset (Zhang et al., 2023b). Left: Visualization of the distribution of the artifacts area ratio, calculated from 1405 annotated artifact images. The histogram demonstrates that generative artifacts are usually *tiny*; Right: Visualization of the artifact masks randomly selected. It demonstrates that most artifacts are tiny and *irregular-shaped*.

Since PAL released a large-scale dataset for artifacts detection containing generated images and segmentation labels, we perform a data analysis on a randomly chosen subset. In Fig. 8, the histogram on the left shows the distribution of the area ratio of artifacts; the figure on the right visualizes the artifact masks sampled from the dataset. The conclusions can be summarized as follows:

- Artifact regions are typically very small, with the artifact area covering less than 4% of the entire image in more than 50% of cases.
- Artifacts exhibit irregular shapes.

The design of our refinement model is motivated by these observations.

### A.3 TRAINING DETAILS

During the training, the whole U-Net, the cross-attention modules and the MLP connecting DINOv2 and the backbone have been fine-tuned. Our training set consists of 1) Pixabay, a dataset of 116k images; 2) MVObj, a dataset of 51k paired images. We train the model with a batch size of 192 and drop the image embedding at a rate of 0.1. The learning rate of the MLP connecting DINOv2 and U-Net is $4 \times 10^{-5}$, and the U-Net has a learning rate of $1 \times 10^{-5}$. The model is trained for more than 45 epochs on 8 NVIDIA A100 GPUs.

### A.4 POST-PROCESSING OF THE CORRESPONDENCE MAP

We apply some post-processing on the raw $M^{t,l}$ to obtain the optimal correspondence map $M^*$ which is clean. As Darcet et al. (2023) pointed out, noise is often identified around the corners and boundaries in feature maps of ViT networks. Since such noise is also observed in our case, we utilize a noise filter to remove it via peak detection. After noise removal, there are only a few blobs left; and we simply adopt a clustering algorithm to locate the largest blob as the corresponding region.

### A.5 USER STUDY

We show the two sections of our user study in Fig. 9 and Fig. 10, measuring realism and fidelity respectively. Note that the images shown in the figures here have been resized for display purposes (thus appearing smaller) and do not reflect their actual sizes used in the user study.

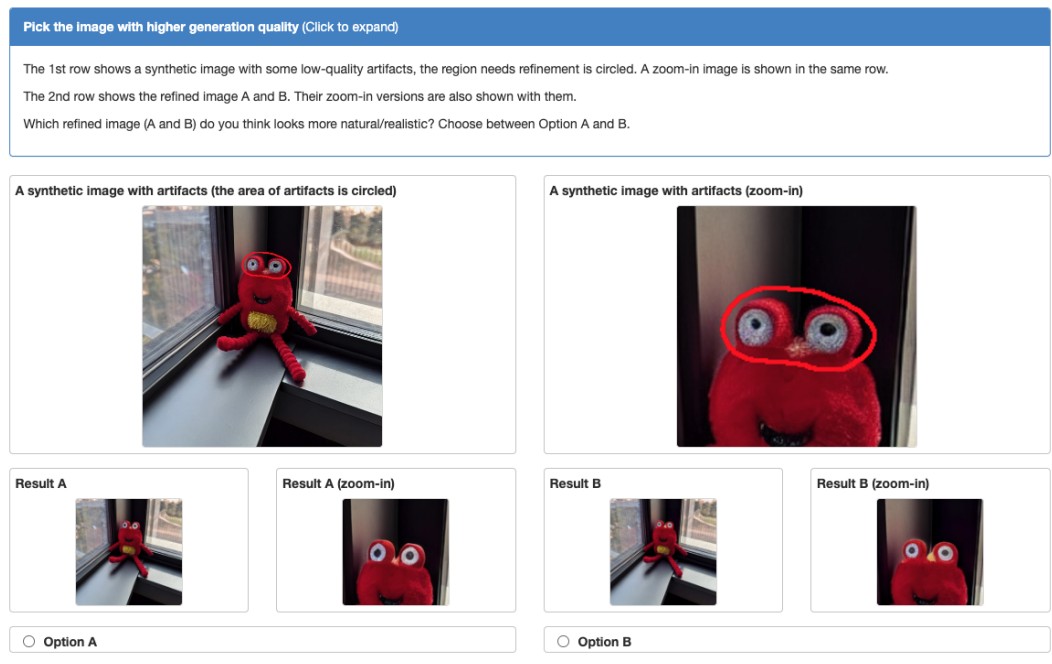

Figure 9: User interface of the user study evaluating the overall quality.

### A.6 GENARTIFACTBENCH

The features of proposed benchmark, *GenArtifactBench*, are listed in Sec. 4.1. We collect 146 images groups for four tasks: Text-to-Image customization, novel view synthesis, object composition and virtual try-on; the synthetic images are generated by DreamBooth (Ruiz et al., 2023a), Zero123++ (Shi et al., 2023b), AnyDoor (Chen et al., 2023) and IDM-VTON Choi et al. (2024). Fig. 11 shows one example for each task.

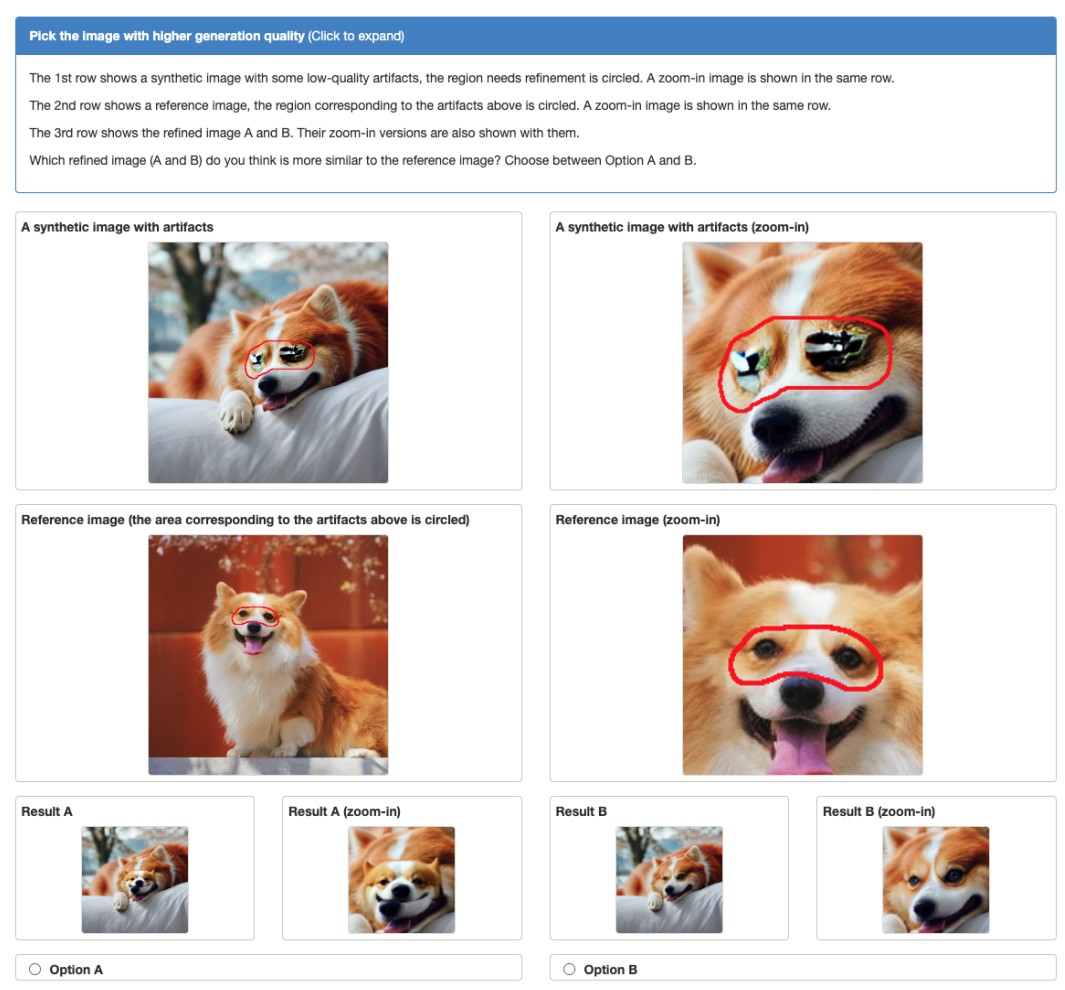

Figure 10: User interface of the user study evaluating identity preservation.

## A.7    ADDITIONAL QUALITATIVE RESULTS

We include more qualitative results in Fig. 12.

## A.8    EXAMPLES OF THE MVOBJ DATASET

As part of our training dataset, we have collected *MVObj*, an object-centric paired dataset. Examples are displayed in Fig. 13.

## A.9    QUALITATIVE STUDY OF THE ROBUSTNESS

In real-world applications, the generated images exhibit significant diversity in structure and appearance, making it essential to evaluate the robustness of our model, especially in cases where artifact regions and reference objects differ substantially in appearance. As shown in Fig. 14, we selected examples where the images with artifacts and the references have a substantial domain gap in content. We then applied our refinement model to locate the correspondences in the references. The results demonstrate the robustness of our model.

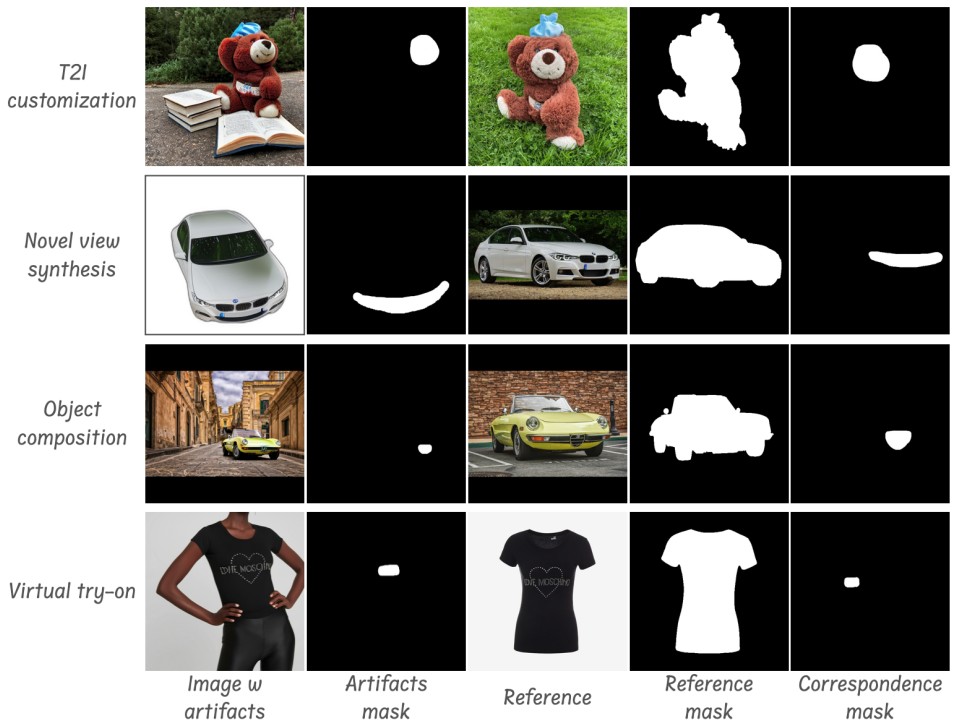

Figure 11: Example images of our proposed benchmark, *GenArtifactBench*.

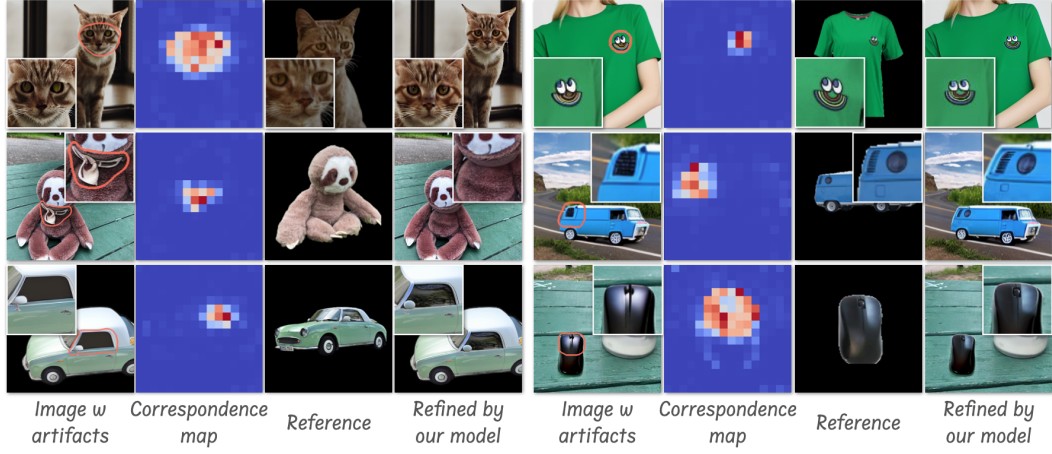

Figure 12: More qualitative results.

## A.10 ADDITIONAL VISUAL RESULTS FOR VIEW SYNTHESIS

We include more qualitative results in Fig. 15, refining novel view synthesis results.

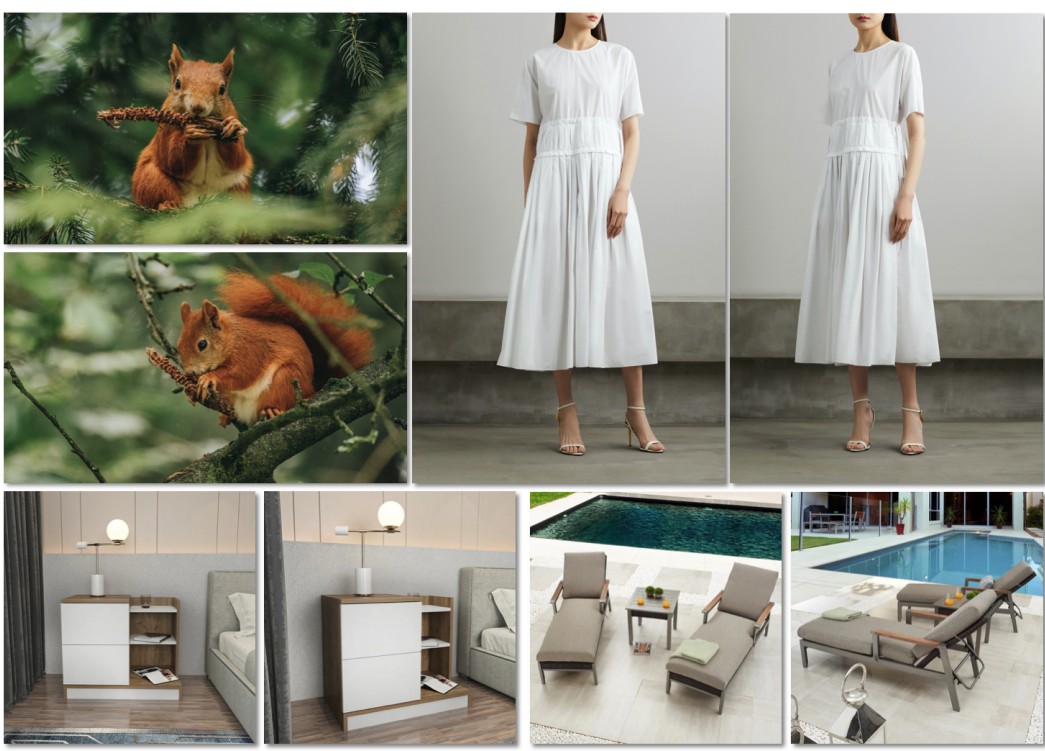

Figure 13: A few paired images of the training dataset MVObj.

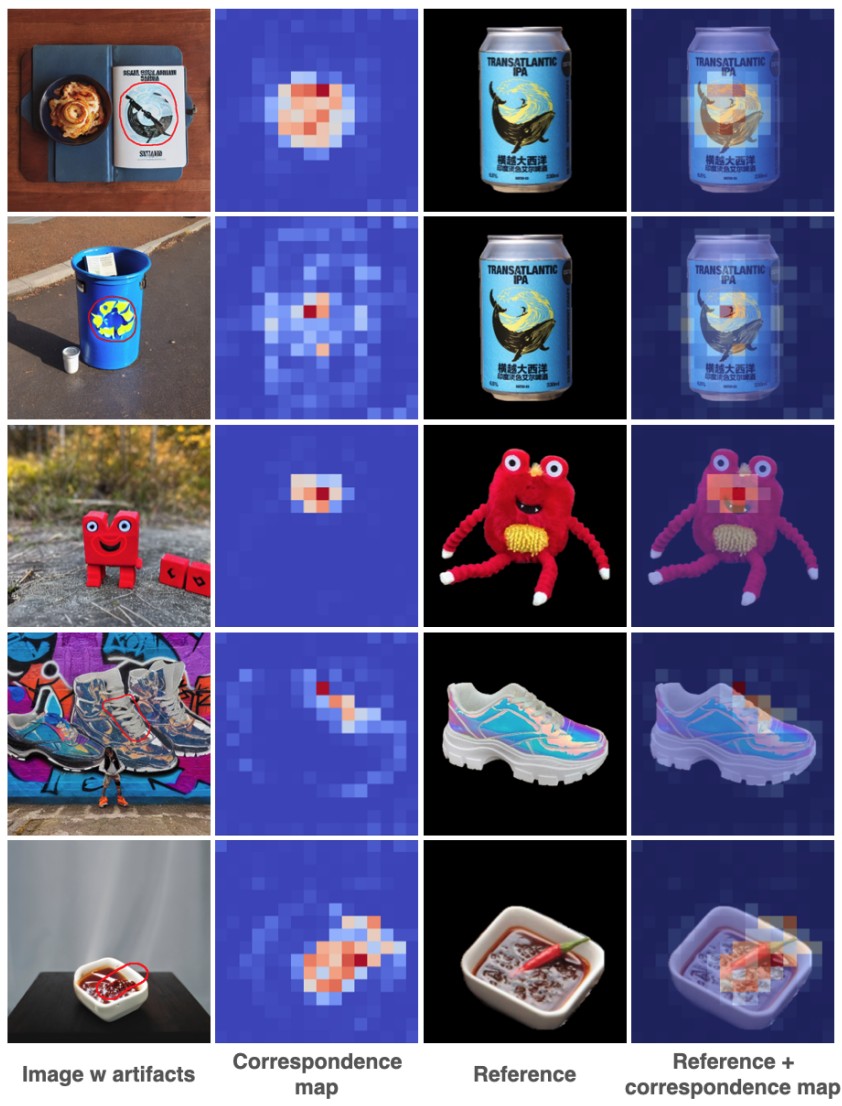

Figure 14: Qualitative analysis of the robustness of the input data. The first column shows the generated images, with artifacts highlighted by red circles. Given the generated images (first column) and the references (third column), our model generates the correspondence maps (second column). Even when the artifact regions and reference objects differ significantly in appearance (e.g., shape, texture, or color), our model is capable of achieving accurate alignment.

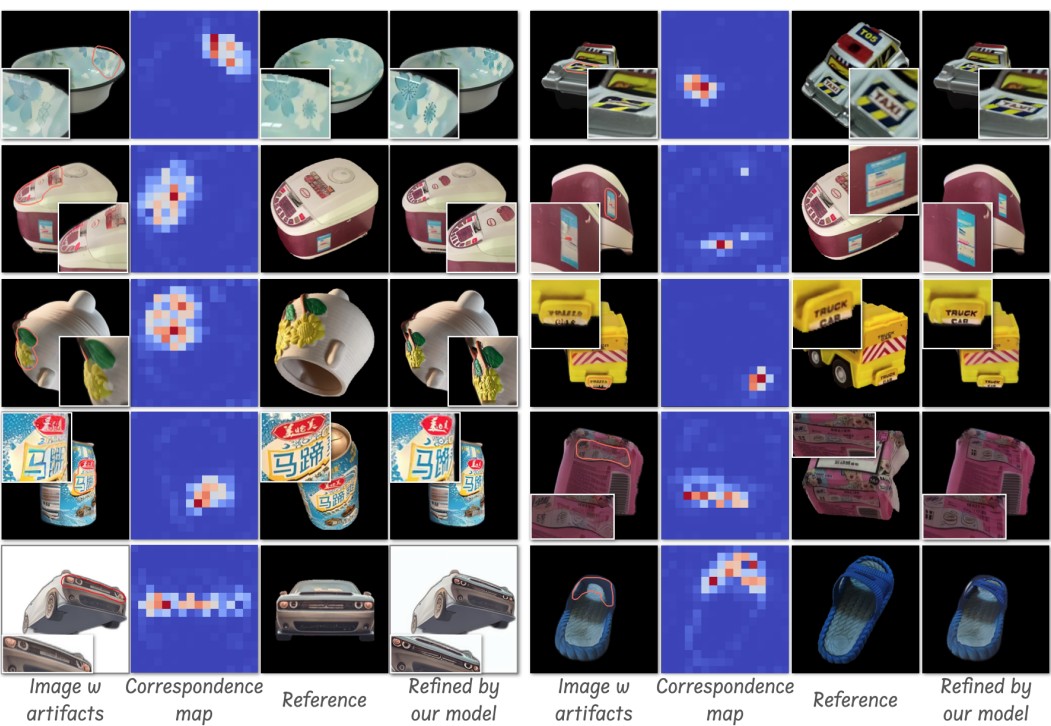

Figure 15: To evaluate the robustness of our model on large view changes between the artifact image and the reference, we further collect a test set based on Zero 1-to-3++ (Shi et al., 2023b) and refine the view-synthesis results.