# OpenReview forum: "Refine-by-Align: Reference-Guided Artifacts Refinement through Semantic Alignment"
_ICLR.cc/2025/Conference — ICLR 2025 Poster_

### Official Review · Reviewer_azdH · 2024-10-25

**Soundness:** 2
**Presentation:** 3
**Contribution:** 3
**Rating:** 6
**Confidence:** 3

**Summary:**

This paper introduces Refine-by-Align, a novel model for reference-guided artifact refinement in personalized image generation. It employs a two-stage diffusion-based framework to enhance image fidelity and identity without requiring test-time tuning, significantly improving fine details across various applications.

**Strengths:**

1.This paper is clearly written and easy to understand.

2.The authors conducted extensive experiments to demonstrate the effectiveness of the method.

3.It is meaningful for the practical applications of customized generation.

**Weaknesses:**

The visual results presented in the paper mostly showcase examples where the target region in the reference image and the corresponding region in the given image with artifacts have little difference in angle and position. The reviewer would like to see the effectiveness across a broader range of tests, which is important for practical applications.

**Questions:**

The reviewer raised concerns about the task workflow design. Since it is necessary to annotate the artifact mask regions in the image with artifacts during inference, why not simultaneously annotate the corresponding mask regions in the reference image, instead of relying on feature matching?

---

> ### Author Response · Authors · 2024-11-25
> **Rebuttal**
>
> We thank the reviewer for careful comments on our work and provide our responses below:
>
> **In this reply, we provide new results for novel view synthesis in Fig. 18.**
>
> > The paper mostly showcases examples where the target region in the reference and the artifacts in the given image have little difference in angle and position.
>
> Please refer to **Fig. 18** in the revised Appendix for the results with larger view changes.
>
> > Why not simultaneously annotate the corresponding mask regions in the reference image, instead of relying on feature matching?
>
> We design this workflow based on the following reasons:
>
> **(1)** Our pipeline is not the only one using this design. E.g., MimicBrush uses exactly the same input format as ours.
>
> **(2)** We try to automate the whole editing process as much as possible to simplify it for the users in the real-world use case. Directly providing the correspondence mask as an additional input is also supported in our pipeline.

---

> ### Author Response · Authors · 2024-11-27
> **A kind reminder**
>
> We sincerely appreciate the time and effort you have dedicated to reviewing our submission. We have submitted our response and would like to follow up to inquire whether our response has addressed your concerns.
>
> Please do not hesitate to let us know if you have any remaining questions or clarification. We look forward to hearing from you. We are eager to address them promptly before the discussion deadline.
>
> Thank you again for your valuable insights and comments.

---

> ### Author Response · Authors · 2024-12-01
> **Kind Reminder**
>
> Hi Reviewer azdH, We sincerely appreciate the efforts you have dedicated to reviewing our submission. We have submitted our **response and new results**, and would like to follow up to inquire whether our response has addressed your concerns.
>
> Please let us know if you have any remaining questions. We look forward to hearing from you and we are eager to address them promptly before the discussion deadline.
>
> Thank you again.

---

> > ### Comment · Reviewer_azdH · 2024-12-02
> >
> > I appreciate the authors’ response and will maintain my score. Thank you.

---

### Official Review · Reviewer_Hbs1 · 2024-10-31

**Soundness:** 3
**Presentation:** 2
**Contribution:** 2
**Rating:** 5
**Confidence:** 4

**Summary:**

This paper proposed a reference-guided artifacts refinement method, which follows the Refine-by-Align. Specifically, the proposed method can be divided into two stages: the alignment stage and the refinement stage. The alignment stage leverages the cross-attention maps to locate the artifact region in DINOv2 features (target view). Then the target view would be filtered by the mask and then refines the final results through the reference-guided inpainting. The authors present a new GenArtifactBench dataset to verify the effectiveness of the proposed method. The proposed method outperforms other competitors in artifact refinement. Moreover, this paper proposed a manually annotated dataset called MVObj for training.

**Strengths:**

1. The newly proposed task, reference-guided artifacts refinement is interesting, which could facilitate many tasks, such as reference-guided editing and novel view synthesis.
2. The solution of Refine-by-Align is reasonable.
3. The manually collected MVObj is interesting, including multi-view scenes with non-rigid objects, such as animals shown in Fig.13. Unfortunately, no more details are discussed to clarify how to collect this dataset in this paper.

**Weaknesses:**

1. The overall typesetting of this paper is confusing. For example, Fig.3, Fig.4, and Fig.5 are displayed in completely unrelated paragraphs. Moreover, the order of Fig3,4,5 is also confusing, while Fig.5 should be mentioned before Fig.4.
2. More details about MVObj should be discussed in the paper.
3. As discussed in the limitation, the proposed method struggles to repair artifacts with large viewpoint changes, which largely restricts its application. Most cases shown in the paper could be addressed as a simple homography warping. More examples of novel view synthesis should be considered as the paper claims.
4. Actually, the experiments in this paper are not solid and fair enough. Artifact removal is a new task proposed in this paper, while all other competitors are not specially trained for this. As shown in Fig.6, all other methods are completely failed.

**Questions:**

1. Will the benchmark and MVObj be open-released?
2. The authors should provide more novel view synthesis examples and clarify it. Otherwise, the claim to address novel view synthesis would conflict with the limitation of "large disparity".
3. It is interesting to explore whether larger target image with more DINOv2 tokens would benefit the final performance.

---

> ### Author Response · Authors · 2024-11-25
> **Rebuttal**
>
> We appreciate the helpful comments from the reviewer and provide our responses below.
>
> **In this reply, we provide new results for view synthesis in Fig. 18, and carefully specify the requested implementation details.**
>
> > The order of Fig.3,4,5 is also confusing, while Fig.5 should be mentioned before Fig.4.
>
> We will swap Fig.4 and Fig.5, and move Fig.3,4,5 to Sec. 3.
>
> > More details about MVObj should be discussed in the paper.
>
> MVObj is an object-centric multi-view dataset, and the images come from the Internet. When there is a sequence of high-quality images which contains the same object captured from multiple views, we will manually choose 2-5 images to form a group of our training set. As a result, we collected more than 50k groups. We then run segmentation and obtain the binary mask of the most prominent object in the image. This dataset covers a wide range of categories, which we classified into animals, art department stores, clothing department stores, digital electronics, furniture appliances, plants and transportation. We display several examples in **Fig. 13**, and will revise the paper with the details above.
>
> > More examples of novel view synthesis should be considered as the paper claims.
>
> Please refer to **Sec. A12** and **Fig. 18** in the revised Appendix.
>
> > The experiments in this paper are not solid and fair enough. Artifact removal is a new task proposed in this paper, while all other competitors are not specially trained for this.
>
> Since we define a new task of reference-guided artifacts refinement, there are not many relevant works. However, we still tried our best to achieve fair comparison:
>
> **(1)** We compare Refine-by-Align with the SDXL version of PAL, which is the most relevant model for artifacts removal.
>
> **(2)** We also evaluate MimicBrush, which has the most similar setting with our pipeline.
>
> **(3)** For the other reference-guided object inpainting models (Paint-by-Example, ObjectStitch), we did not use their original versions; instead, we **adapted them for our task** by tuning their checkpoints on our training data. We will revise our paper and explain that we used the adapted versions as the baselines.
>
> > Will the benchmark and MVObj be open-released?
>
> Currently we do not have a plan to release MVObj (it is not claimed as a contribution either); but we will fully release the benchmark upon acceptance.
>
> > The authors should provide more novel view synthesis examples.
>
> Please refer to **Fig. 18** in the revised Appendix.
>
> > It is interesting to explore whether a larger target image with more DINOv2 tokens would benefit the final performance.
>
> We would like to thank the reviewer for this advice. It is a good idea but with the following trade-off:
>
> **Pros:** The image encode can encode more low-level details from the reference image, and they are better preserved in the refined image. We can simply increase the input resolution from 224x224 to 448x448, and double the number of the image tokens.
>
> **Cons:** It will increase both the memory cost and the inference time.

---

> ### Author Response · Authors · 2024-11-27
> **A kind reminder**
>
> We sincerely appreciate the time and effort you have dedicated to reviewing our submission. We have submitted our response and would like to follow up to inquire whether our response has addressed your concerns.
>
> Please do not hesitate to let us know if you have any remaining questions or clarification. We look forward to hearing from you. We are eager to address them promptly before the discussion deadline.
>
> Thank you again for your valuable insights and comments.

---

> ### Author Response · Authors · 2024-12-01
> **Kind Reminder**
>
> Hi Reviewer Hbs1,
> We sincerely appreciate the efforts you have dedicated to reviewing our submission. We have submitted our **response and new results**, and would like to follow up to inquire whether our response has addressed your concerns.
>
> Please let us know if you have any remaining questions. We look forward to hearing from you and we are eager to address them promptly before the discussion deadline.
>
> Thank you again.

---

### Official Review · Reviewer_bn5t · 2024-11-03

**Soundness:** 3
**Presentation:** 3
**Contribution:** 3
**Rating:** 6
**Confidence:** 3

**Summary:**

This paper addresses an intriguing problem where a guidance image is used to restore artifacts in local regions of generated images. To achieve this, a two-stage approach is proposed: (i) an alignment stage that extracts the target region from the reference by computing correlation with the input ROI content; and (ii) a refinement stage that utilizes features from the reference patch to address the artifacts in the input ROI. Experiments demonstrate the superiority of the proposed method over several baselines, and ablation studies validate the importance of the design choices.

**Strengths:**

- The proposed task is well-motivated and interesting, potentially benefiting practical scenarios in reference-guided image generation applications.

- The idea of the proposed two-stage pipeline is simple yet effective, contributing valuable insights to the community.

- The experiments are comprehensive, effectively justifying the major claims and the significance of the proposed innovation.

**Weaknesses:**

- As claimed in the paper, the proposed method is model-agnostic and applicable to diverse tasks. Therefore, it is crucial to study the robustness of the input data. Although stated as a limitation, a comprehensive study is still needed to show the milestones the proposed method has achieved and what remains for future work.

	- For the alignment stage, will the performance degrade if the input artifact images contain different contents (though still sharing the common object) compared to the reference image?
	- For the refinement stage, will the performance degrade if the ROI of the input artifact images is not very similar to that in the reference image (possibly due to structural distortion from image-guided generation algorithms)?

- As the major contribution, the importance of the two-stage method is validated in a less convincing manner. In Table 3, the metrics may not effectively capture the local artifacts. It would be stronger if a user study were provided, i.e., a comparison between the full method and the model trained only with the alignment mode using the full reference image directly.

**Questions:**

- The training scheme should be clarified more clearly. In its current form, it is unclear which parts of the model will be trained. For example, if the model of Anydoor is adopted, will the full trainable parameters of Anydoor and the newly added modules by the proposed method be trained jointly? If so, how do you avoid the issue of overfitting or damaging the capability of the pre-trained Anydoor when the U-Net is trained on the collected dataset?

- For the proposed task, one key consideration should be the generalization ability to diverse artifacts induced by different task-specific models. Therefore, it is crucial to design a comprehensive image degradation simulation pipeline. Can you elaborate on the data perturbation used in preparing the training dataset?

---

> ### Author Response · Authors · 2024-11-25
> **Rebuttal**
>
> We thank the reviewer for the helpful comments and provide our responses below:
>
> **In this reply, we add several new experiments: Fig. 15 as an evaluation of the model robustness, an additional user study to validate our two-stage pipeline, and Fig. 16,17 as a degradation simulation pipeline. We also provide more implementation details.**
>
> > For the alignment stage, will the performance degrade if the input artifact images contain different contents (though still sharing the common object) compared to the reference image?
>
> Please refer to **Sec. A.10** and **Fig. 15** in the revised Appendix. We show the results of correspondence matching when the input artifacts image and the reference have very different appearances. Our alignment model will still work under this case, since it finds correspondences based on semantic features, not just from low-level features. This is accomplished by the training strategy under the alignment mode (see **Sec. 3.4**).
>
> > For the refinement stage, will the performance degrade if the ROI of the input artifact images is not very similar to that in the reference image (possibly due to structural distortion from image-guided generation algorithms)?
>
> There are two cases:
>
> **(1)** If the domain gap in **color** is large (e.g., the input object with artifacts is blue but the reference object is red), then the refined results may look unrealistic around the transition area. During the training, the color perturbation we introduced is within a specific range (10%), so the model’s ability to change color is limited. However, the texture details can still be improved.
>
> **(2)** If the domain gap in **structure** within the ROI is large, then the performance will not degrade. The ROI will simply be replaced by the reference (see the bottom-left in **Fig. 1**, the last example of **Fig. 6**, and the middle row of **Fig. 12**).
>
> > The importance of the two-stage method is validated in a less convincing manner. It would be stronger if a user study were provided, i.e., a comparison between the full method and the model trained only with the alignment mode using the full reference image directly.
>
> We conducted an additional user study as suggested and the results below show the user preference (in percentage):
>
> |     | Identity ↑| Quality ↑|
> | --- | --- | --- |
> | One stage  | 44.17 | 38.33 |
> | Full model | **55.83** | **61.67** |
>
> Where we designed two problems to measure identity preservation and realism. It further demonstrates the effectiveness of our two-stage method.
>
> > It is unclear which parts of the model are trained. How do you avoid the issue of overfitting or damaging the capability of the pre-trained Anydoor when the U-Net is trained on the collected dataset?
>
> We finetuned the whole U-Net, the cross-attention modules and the MLP connecting DINOv2 and the backbone. We will revise our paper and add these details. Actually we find the original AnyDoor checkpoint tends to overfit by copy-and-pasting the reference onto the artifacts region (see **Fig. 6**). To resolve this issue, we introduce color and spatial perturbations to our single image dataset (Pixabay) and also train the model with a large multi-view dataset (MVObj, see **Fig. 13** for examples). We did not observe overfitting during our training.
>
> > It is crucial to design a comprehensive image degradation simulation pipeline. Can you elaborate on the data perturbation used in preparing the training dataset?
>
> We would like to thank the reviewer for this advice. Currently we use color and spatial/perspective perturbation in our training data, and also include a multi-view dataset containing natural perturbations. Following this advice, we further design a pipeline to simulate the artifacts in **Sec. A.11** and **Fig. 16** of the revised Appendix. Example training pairs generated by this pipeline can be found in **Fig. 17**.

---

> > ### Comment · Reviewer_bn5t · 2024-11-26
> >
> > Thanks for the updated results. I have no more questions and will maintain my current rating.

---

### Official Review · Reviewer_FAkR · 2024-11-06

**Soundness:** 3
**Presentation:** 2
**Contribution:** 2
**Rating:** 6
**Confidence:** 2

**Summary:**

The paper proposes an artifacts refinement framework to refine the artifacts (e.g., logos, details) in the generated image by leveraging the corresponding details from a reference image. The method is evaluated qualitatively and quantitatively on a new benchmark, GenArtifactBench, consisting of artifacts generated by several well-known models, reference images, and dense human annotations.

**Strengths:**

1.The visual results are interesting and show the effectiveness of the method for fixing different types of details.

2.The proposed benchmark could be a good complementary dataset to facilitate studies on the reference-based image inpainting.

**Weaknesses:**

1.I feel like the overall pipeline is still quite similar to previous reference-based image inpainting methods (Paint-by-Example, ObjectStitch). Using cross attention to localize matches is also already prevalent in the community. So the technical novelty is a bit limited. Could the authors please explain more why this is more effective?  What is the component that distinguishes the method from previous ones? Is it mainly because of the zoom-in of the reference image?

2.The method focuses on the details of the reference-based inpainting results, I wonder if there are any advantages of the method given an object level inpainting? If so, why? If not, then I feel the scope of the method may not be not general enough.

3.For evaluation, I wonder if it is possible to also evaluate the CLIP score on the masked region only, which potentially may be a better metric for the task.

**Questions:**

I think the paper adds values to better reference-based inpainting literature, but the significance of the scope and technical differences may require further explanation.

---

> ### Author Response · Authors · 2024-11-25
> **Rebuttal (part 1/2)**
>
> We thank the reviewer for the insightful comments and provide our responses to the reviewer's questions one by one below.
>
> **To summarize, our method significantly differs from the reference-based inpainting methods (proved by an additional experiment in Fig. 14) and other feature matching methods with cross attention; our model can be applied on top of any generative models, including T2I personalization, virtual try-on and 3D tasks such as view synthesis.**
>
> > The overall pipeline is still quite similar to previous reference-based image inpainting methods (Paint-by-Example, ObjectStitch). Using cross attention to localize matches is also already prevalent in the community. Could the authors please explain more why this is more effective? What is the component that distinguishes the method from previous ones?
>
> There are two components that distinguish our method from the previous works that contribute to the effectiveness:
>
> **(1)** Removing the irrelevant information from the reference via correspondence matching. As proved by the first row of **Fig. 3**, using a cropped local area from the reference can significantly improve the fidelity and identity.
>
> **(2)** The zoom-in strategy. As verified by **Fig. 10** in Perceptual Artifacts Localization (PAL, Zhang et al., 2022), images generated using zoom-in have better quality and less artifacts.
>
> \
> The main differences between our method and reference-based inpainting methods are as follows:
>
> **(1)** Our model focuses on a task that is totally different with the previous reference-based image editing. While these methods aim to insert or blend an object into a background, our method aims at a general improvement for any generated images. In other words, our method can be built on top of these methods.
>
> **(2)** Paint-By-Example (PbE), ObjectStitch (OS) and IMPRINT are **unable** to perform semantic alignment. They all start by converting the reference to 256 patch tokens (which contains the spatial information), but then either discard the patch tokens (PbE) or reduce 256 to 77 (OS and IMPRINT). As a result, the spatial information has been lost in the reference embeddings. However, in our model, we preserve the spatial information by feeding all 256 patch tokens to the cross attention layers.
>
> **(3)** Although AnyDoor can perform semantic alignment, the accuracy is much worse than our model, since our model has been tuned with carefully-designed training pairs (see **Sec 3.4**) to improve the correspondence-matching ability. Additional results of using AnyDoor for the alignment are shown in **Sec. A.9** and **Fig. 14** of the revised Appendix, demonstrating the weakness of AnyDoor in alignment.
>
> \
> To the best of our knowledge, methods using cross attention for feature matching are not very common, as there are recently only two related works: Cross Image Attention (Alaluf et al., 2023) and LDM Correspondences (Hedlin et al., 2023). The latest methods for correspondence matching includes DIFT (Tang et al., 2023), Diffusion Hyperfeatures (Luo et al., 2023) and A Tale of Two Features (Zhang et al., 2023), where the intermediate diffusion features are mainly leveraged for semantic matching. Nonetheless, our method still differs from Cross Image Attention and LDM Correspondences in the following aspects:
>
> **(1)** Application scope. Cross Image Attention and LDM Correspondences are mainly used for key-point matching, while our method is mainly aiming at free-form region matching (see **Fig. 2**).
>
> **(2)** Efficiency. Although Cross Image Attention is zero-shot, the correspondence matching has to be repeated across multiple timesteps. In LDM Correspondences, the embeddings need to be optimized before inference. In contrast, our method only takes 1 denoising timestep to find the correspondence.
>
> **(3)** Architecture. We calculate the cross attention between the latent features and DINOv2 embedding; in LDM Correspondences, this operation is between the latent features and the text embedding; in Cross Image Attention, the operation is between two latent features and additional efforts such as AdaIN and a contrast operation.
>
> **(4)** Our method greatly outperforms Cross Image Attention, as shown in **Tab. 1, Tab. 2** and **Fig. 6**.

---

> ### Author Response · Authors · 2024-11-25
> **Rebuttal (part 2/2)**
>
> > I wonder if there are any advantages of the method given an object level inpainting? If so, why? If not, then I feel the scope of the method may not be not general enough.
>
> We leverage a reference-based inpainting framework as our basic architecture, so that we can take advantage of a pretrained checkpoint (e.g., AnyDoor) and the post-training is easier to converge. Another advantage is that we do not have to compute the correspondence or train it from scratch, since the spatial correspondence is already contained in the cross-attention layers of such frameworks and we only extract this information (inspired by Prompt-to-Prompt, Hertz et al., 2022).
> The application scope of our method is not reference-guided inpainting, it can be applied on **the results of any general image generation task**. E.g., **Fig. 1, Fig. 6** also show extensive examples of applying our model on text-to-image personalization, virtual try-on and 3D tasks such as novel view synthesis.
>
> > For evaluation, I wonder if it is possible to also evaluate the CLIP score on the masked region only, which potentially may be a better metric for the task.
>
> Although not specified in the paper, the current results are already evaluated **only over the masked region**. We will add this information in the revised paper.

---

> ### Author Response · Authors · 2024-11-27
> **A kind reminder**
>
> We sincerely appreciate the time and effort you have dedicated to reviewing our submission. We have submitted our response and would like to follow up to inquire whether our response has addressed your concerns.
>
> Please do not hesitate to let us know if you have any remaining questions or clarification. We look forward to hearing from you. We are eager to address them promptly before the discussion deadline.
>
> Thank you again for your valuable insights and comments.

---

> ### Author Response · Authors · 2024-12-01
> **Kind Reminder**
>
> Hi Reviewer FAkR,
> We sincerely appreciate the efforts you have dedicated to reviewing our submission. We have submitted our **response and new results**, and would like to follow up to inquire whether our response has addressed your concerns.
>
> Please let us know if you have any remaining questions. We look forward to hearing from you and we are eager to address them promptly before the discussion deadline.
>
> Thank you again.

---

### Meta-Review · Area_Chair_wdue · 2024-12-11

**Metareview:**

The paper receives mixed but mostly positive reviews by the reviewers after the rebuttal. Reviewers appreciate the interesting task, the simple yet effective design, and the extensive experiments. Reviewer Hbs1 raised several concerns regarding the details of MVObj, missing novel view examples, and unfair comparisons. However, most of them are addressed in the authors' rebuttal. AC did not find any major remaining concerns to reject the paper and agrees with other reviewers.

**Additional Comments On Reviewer Discussion:**

During the discussion, reviewer Hbs1 did not respond to the rebuttal or express new opinions. AC checked the rebuttal and agrees the major concerns have been addressed by the authors' response.

---

### Decision · Program_Chairs · 2025-01-22

Accept (Poster)